# Cytokine Profile in Lung Cancer Patients: Anti-Tumor and Oncogenic Cytokines

**DOI:** 10.3390/cancers15225383

**Published:** 2023-11-13

**Authors:** Freddy Elad Essogmo, Angelina V. Zhilenkova, Yvan Sinclair Ngaha Tchawe, Abah Moses Owoicho, Alexander S. Rusanov, Alexander Boroda, Yuliya N. Pirogova, Zaiana D. Sangadzhieva, Varvara D. Sanikovich, Nikolay N. Bagmet, Marina I. Sekacheva

**Affiliations:** 1Institute for Personalized Oncology, Center for Digital Biodesign and Personalized Healthcare, First Moscow State Medical University of the Ministry of Health of Russia (Sechenov University), Moscow 119991, Russia; eladfreddy96@yahoo.com (F.E.E.); av.zhilenkova@gmail.com (A.V.Z.); ngahayvan@gmail.com (Y.S.N.T.); abahmoses28@gmail.com (A.M.O.); alexrus146@yandex.ru (A.S.R.); alexcom90@yandex.ru (A.B.); pirogova_yuliya96@mail.ru (Y.N.P.); sangadzhieva.md@gmal.com (Z.D.S.); v8varvara@gmail.com (V.D.S.); 2Cameroon Oncology Center (COC), Douala P.O. Box 1864, Cameroon; 3Petrovsky National Research Centre of Surgery, Moscow 117418, Russia; bagmetn@mail.ru

**Keywords:** lung cancer, cytokines, therapy, immune system, anti-tumor

## Abstract

**Simple Summary:**

Lung cancer is currently the second leading cause of cancer death worldwide. Cytokines are small proteins that carry messages between cells and are known to play an important role in the body’s response to inflammation and infection. Cytokines are important for immunity in lung cancer. The immune system relies heavily on cytokines which can also be produced in the laboratory for therapeutic use. Cytokine therapy helps the immune system to stop the growth or kill cancer cells including lung cancer cells. High doses of cytokines are required to induce a beneficial response in cancer patients, but doing so leads to many problems, including their short lifespan and toxicity. New technologies are being developed to help improve the targeting of cytokines and alter their side effects.

**Abstract:**

Lung cancer is currently the second leading cause of cancer death worldwide. In recent years, checkpoint inhibitor immunotherapy (ICI) has emerged as a new treatment. A better understanding of the tumor microenvironment (TMJ) or the immune system surrounding the tumor is needed. Cytokines are small proteins that carry messages between cells and are known to play an important role in the body’s response to inflammation and infection. Cytokines are important for immunity in lung cancer. They promote tumor growth (oncogenic cytokines) or inhibit tumor growth (anti-tumour cytokines) by controlling signaling pathways for growth, proliferation, metastasis, and apoptosis. The immune system relies heavily on cytokines. They can also be produced in the laboratory for therapeutic use. Cytokine therapy helps the immune system to stop the growth or kill cancer cells. Interleukins and interferons are the two types of cytokines used to treat cancer. This article begins by addressing the role of the TMJ and its components in lung cancer. This review also highlights the functions of various cytokines such as interleukins (IL), transforming growth factor (TGF), and tumor necrosis factor (TNF).

## 1. Introduction

Lung carcinoma, also known as lung cancer, is a type of lung cancer that usually results from uncontrolled cell proliferation of the lungs. Tissues containing epithelial cells that have transformed into malignant cells are examples of lung cancer. Abnormal masses or nodules may be seen on chest X-rays [1]. A computerized tomography (CT) scan can also help to see small lung lesions that X-rays may have missed.

Lung cancer is currently the second leading cause of cancer worldwide. Although lung cancer can occur in people who have never smoked, people who smoke are the group most likely to develop lung cancer. Both the amount and frequency of smoking can affect a person’s risk of developing lung cancer. People who quit smoking after smoking for a long time will have a decreased risk of developing lung cancer [1].

To reduce the number of deaths from lung cancer, the best option is to increase the effectiveness of cancer prevention and to use screening strategies for risk assessment and detection in the early stages of lung cancer treatment [2]. According to a study by R. Kuner, the estimated number of new cases and deaths from lung cancer in the United States in 2013 was 228,190 and 159,480, respectively. Approximately 55% to 60% of patients have distant metastases and are diagnosed at an advanced, incurable stage. Therefore, the five-year overall survival rate of each stage is between 13% and 16% [3].

Non-small cell lung cancer, also known as NSCLC, is the most common type of bronchial malignancy. Adenocarcinoma and squamous cell carcinoma are the two main histological groups from which they are often separated. Adenocarcinoma and squamous carcinoma cells differ in DNA copy number, DNA methylation, gene mutations, transcriptome, proteome, and potential biomarkers [2].

In lung cancer, cytokines, or proteins, are important to support the immune system. They can promote tumor growth (oncogenic cytokines) or inhibit tumor growth (anti-tumor cytokines) by modulating related factors such as growth, proliferation, metastasis, and apoptosis [2].

### 1.1. Lung Cancer

The lungs are the foundational organs of the respiratory system, whose most basic function is to facilitate gas exchange from the environment into the bloodstream. Oxygen is transported through the alveoli into the capillary network, where it can enter the arterial system and ultimately perfuse the tissue. The respiratory system is composed primarily of the nose, oropharynx, larynx, trachea, bronchi, bronchioles, and lungs. The lungs further divide into individual lobes, which ultimately subdivide into over 300 million alveoli. The alveoli are the primary location for gas exchange [4]. Lung cancer is divided into small cell lung cancer (SCLC), which accounts for 15% of patients, and non-small cell lung cancer (NSCLC), which accounts for 85% of patients [5]. Histologically, NSCLC is divided into three subtypes: adenocarcinoma (ADC), large cell lung cancer (LCC), and squamous cell carcinoma (SCC) [6]. Both smokers and nonsmokers have been found to have NSCLC, which is the most common variety that is gender-neutral, affecting 40% of ADC victims [7]. ADC is made up of type II alveolar cells in the outer lung, which are mucus sealing cells [8]. Other subtypes of SCC occur in flattened squamous cells in the airways, centrally in the lungs, and account for approximately 25% to 30% of the most common causes of smoking [6].

Some asymptomatic cancer patients cannot be accurately diagnosed during conventional treatment. However, approximately 70% of NSCLC patients progress to an advanced stage after diagnosis, either locally or in organ metastases [9]. In other words, early warning signs may be the key to improving cancer patient survival. An important part of the immune system’s specific response to dangerous stimuli is inflammation. Many studies have demonstrated the role of inflammation in cancer [10].

Due to the disease’s complexity, it develops genetic and epigenetic variants. Tumor differentiation, growth, invasion, and metastasis are generally controlled by these modifications. Approximately 70% to 80% of patients do not believe that surgery, which is considered the best treatment for early-stage of NSCLC, is the best option, especially because of locoregional tumor swelling, extrathoracic spread, or a poor physical and functional state at the time of diagnosis. Therefore, patients often choose radiotherapy or chemotherapy alone to achieve a better prognosis [11].

Alternatively, surgical resection follows the neoadjuvant approach. Radiotherapy (RT) is important in the treatment of lung cancer. According to various studies, approximately 60% of patients receive radiation therapy in the early stages of the disease, approximately 44% at diagnosis, and approximately 16% in advanced or relapsed states [12]. Most Food and Drug Administration (FDA)-approved drugs focus on angiogenesis and immunosuppression to demonstrate the biological and physiological dynamics of the tumor microenvironment and cancer-related symptoms [13].

### 1.2. Tumor Microenvironment and Content: Their Contribution to NSCLC Metastasis

Tumors create the tumor microenvironment, which is dominated by interactions brought on by the tumor. The anti-tumor potential of different immune effector cells is down-regulated at the tumor site, although they are usually selected in response to signals from the tumor. The chronic infiltration of inflammatory cells seen in human malignancies is enriched with regulatory T cells (Treg) and myeloid suppressor cells (MSC) [14]. The lymphocytes, macrophages, dendritic cells, and natural killer cells that make up the immune system communicate through cells and other mediators (cytokines and chemokines) [15]. Innate immunity typically works ahead of schedule during the course of an insusceptible reaction and includes a specific configuration of receptors, with the help of macrophages, neutrophils, and NK cells [16].

T lymphocytes (CD4+ and CD8+ T cells) and B cells, on the other hand, regulate adaptive immunity. These cells are prepared by antigen-presenting dendritic cells. B cells are activated by antigens entering their local environment, and CD8+ and CD4+ lymphocytes are sensitive to antigens presented as peptides that form complexes with MHC class I and class II particles, respectively [17]. Germinal gene sequences that produce a large population of T-cell and B-cell receptors can sensitively detect multiple targets, resulting in an immune system that takes days to survive, causing strong memory [18].

The immune response is determined by the cytokine profile transmitted by the immune system. Cell-mediated immunity is favored by TH1 cytokines such as IL-2, IFN-, and tumor necrosis factor (TNF). TH2 cytokines (e.g., IL-4, 5, 10 and 13) are important for immunity and well-being. In addition, TH17 cytokines such as IL-17, 22, and granulocyte sedimentation enhancing component (G-CSF) act to elicit responses [19]. These responses indicate that effector and focused memory T helper type 1 (Th1) and T killer type 1 (Tc1) cells, rather than a humoral immune response, are responsible for defensive long-range resistance against cancer and the ability to combat chronic inflammation [20].

Additionally, antibodies are effective in inhibiting tumor growth when they bind to oncogenic growth receptors such as HER2/NEU and EGFR, and they prefer NK or Fc receptors that activate macrophages or complement protein cascades [21]. When cancer cells metastasize, they leave their original site and live elsewhere, where they can continue to grow [22]. The tumor and its surrounding ME initiate the process of infiltration, forming, and dividing into different tissues, leading to metastasis. The transformation of a normal cell into a tumor is dependent on the environment; therefore, it may occur in one TME but not in another [23]. The TME contains both cellular and non-cellular components (Figure 1) [20].

Lung cancer has a complex process of growth, initiation, and metastasis. Genetic abnormalities acquired by the tumor lead to the development of the disease and interaction with the immune system, leading to a localized ME [24]. Immobility and flexibility in TME involve different mechanisms. First, the TME is the main defense mechanism against foreign invaders and consists of dendritic cells (DC) (CD1c (+), natural killer (NK) cells (CD49a, CD69 and CD103), macrophages (CD68+), NK-T cells CD56+ and CD3+), and neutrophil phagocyte (NK) cells. Cancer cells promote tumor growth, angiogenesis, and metastasis [20].

If the system is reprogrammed, it can also cause tumors to grow. Tumor growth slows from the second phase, which includes B cells (CD20+) and two subsets of T cells, T helper cells (CD4+), and cytotoxic T cells (CD8+) [25]. Tumor infiltrating leukocytes (TILs), which promote anti-tumor responses, account for 67% of TME in lung tumors. This is followed by tumor-associated macrophages (TAMs), followed by small numbers of DC and NK cells [20].

## 2. Basic Properties of Cytokines

The balance of the immune system is controlled by cytokines, which are membrane-bound or released proteins that mediate intercellular signals. They are produced by innate and adaptive immune cells in response to tumor antigens and pathogens. Each cytokine has a unique effect on the immune system that is dependent on many factors, including local cytokine concentrations, cytokine receptor expression patterns, and the integration of various pathways into the immune response. The importance of cytokines in tumor immunity is demonstrated by increased tumor frequency in mice lacking type I or type II interferon (IFN) receptors or downstream IFN receptor signaling components [26].

An important aspect of cytokine signaling is pleiotropy, or the ability of a single cytokine to cause different cell types to produce different effects, some of which may lead to resistance [27] (Table 1). This has been viewed as one of the main challenges of IL-2 therapy due to the dual role of IL-2 as a potent activator of the T regulatory and T effector regions. Another important aspect of cytokine signaling is redundancy, or the number of cytokines with the same function. This reactivity makes it difficult to change cytokines for therapy, because changing one cytokine may induce the other to compensate [26].

Cytokines play complex and often antagonistic roles in immune maturation, host defense, and tumor immunobiology (Table 2). Therefore, the development of cytokine-based immunotherapies for cancer treatment depends on knowing the biological activities and mechanisms of action of these agents [26].

Depending on how they are affected, macrophages can be divided into two types: the active M1 type and activate M2 type. The M1 type secretes Th1 cytokines with pro-inflammatory and anti-tumorigenic properties, while the M2 type secretes Th2 cytokines with anti-inflammatory and pro-tumorigenic properties. Tumor grade and stage correlate with the Th1:Th2 ratio [28,29]

### 2.1. Classification of Cytokines and Their Receptors

To date, seven families of cytokine receptors have been identified [30] (Table 3): type I and type II cytokine receptors, immunoglobulin superfamily receptors, tumor necrosis factor (TNF) receptors, G protein-coupled receptors, transforming growth factor beta (TGF-β), and the recently discovered IL-17 receptor. As these hold immediate clinical promise, this section will focus on cytokines that signal through the type I and type II cytokine receptor families [26].

#### 2.1.1. Type I Cytokine Receptors

Type I cytokine receptors include IL-2, IL-4, IL-7, IL-9, IL-15, and IL-21 receptors and a common chain (c). This chain contains specific Janus kinases (The cytokine component of JAK) 1 and 3 that initiate intracellular signaling via coordination of signal transduction and activation of T (STAT) molecules [31,32,33] (Table 3). Other type I cytokine receptor subsets include the IL-6 and GM-CSF receptor families, which cooperate through the gp130 receptor to influence multiple signaling pathways at their targets [34,35].

Various receptor complexes, such as IL-6, IL-11, leukemia inhibitory factor (LIF), oncostatin M, cardiotrophin-1, and ciliary neurotrophic factor, use gp130 signaling components to influence immunological, hematopoietic excess activity, and pleiotropic effects [36,37,38]. Similarly, receptors of the distinct GM-CSF receptor subfamily, which has a common β chain that binds with the cytokine-specific chain, also recognize IL-3, IL-5, and GM-CSF [39,40].

#### 2.1.2. Type II Cytokine Receptors

Through the use of Type II cytokine receptors, which are made up of a signaling chain and a ligand-binding chain, IFN-α, IFN-β, IFN-γ, and IL-10 actions are mediated. The intracellular domains of type II cytokine receptors frequently interact with Janus kinase (JAK) family tyrosine kinases, whose sequences resemble tandem Ig-like domains [32,41].

#### 2.1.3. Immunoglobulin Superfamily Receptors

IL-1, IL-18, stem cell factor, and monocyte colony stimulating factor receptors are members of the immunoglobulin superfamily and have extracellular immunoglobulin domains [26].

## 3. The Role of Cytokines in Immunotherapy and Cancer

Several cytokines restrict tumor cell growth through direct anti-proliferative or pro-apoptotic activites, or indirectly by stimulating the cytotoxic activity of immune cells against tumor cells. An example is interferon-alpha (IFN-α), which was first discovered in 1957 as a result of its antiviral properties. Thirteen years later, Gresser and Bourali described the anti-tumor activity of IFN-α against different tumor cell lines inoculated in mice [42]. The biggest challenge in the development of anti-cancer drugs is to target and kill only the cancer cells without harming the normal body cells. Interferons (IFNs) are naturally produced by our body cells in response to pathological compromise, and these chemical messengers render the neighboring normal cells resistant to similar types of infection. Interferons regulate angiogenesis and have immunomodulatory capacities, and are hence a fantastic therapeutic choice against cancer [43]. Literature data have demonstrated that inhalation of asbestiform fibers can bring about various inter-connected pathogenetic lung diseases, represented by chronic inflammation and carcinogenesis. Previous investigations have analyzed fundamental pathologic changes following asbestos exposure, highlighting critical pathogenic pathways involving oxidative stress, apoptosis, and inflammation. In 1987, the International Agency for Research on Cancer (IARC) classified asbestos as a group 1 (definite) human carcinogen [44].

i.IL-1:

The interleukin-1 (IL-1) family is one of the first described cytokine families and is made up of eight cytokines (IL-1β, IL-1α, IL-18, IL-33, IL-36α, IL-36β, IL-36γ, and IL-37) and three receptor antagonists (IL-1Ra, IL-36Ra, and IL-38). The IL-1 family members are known to play an important role in inflammation [45]. IL-1 is an important regulator in innate immunity, which can stimulate IFN-γ production by T cells and NK cells. IL-1 plays a dual role in the anticancer immune response. Clinically, patients with high IL-1 concentrations in tumors have poor prognoses [46]. IL-1β is a key mediator in the initiation of the inflammatory response in pulmonary diseases, including chronic obstructive pulmonary disease and lung cancer. Lung inflammation is characterized by macrophage infiltration and increased thickness and fibrosis of the airways, leading to airflow obstruction. IL-1β polymorphisms that lead to IL-1β upregulation may influence the level of reactive oxygen and nitrogen species in the lung epithelial microenvironment, which may invoke inflammatory-mediated induction of mutations in tumor suppressor genes such as TP53. Several strategies are being used to inhibit IL-1 signaling in human disease, including antibodies directed against IL-1α, IL-1β, and the IL-1 receptor. Anakinra, a recombinant version of the naturally occurring IL-1 receptor antagonist, is approved for the treatment of rheumatoid arthritis and cryopyrin-associated periodic syndromes (CAPS) [47].

ii.IL-2:

Interleukin-2 (IL-2), also known as aldesleukin or PROLEUKIN^®^, is considered to be an immunotherapy treatment for people with metastatic melanoma. IL-2 is a naturally occurring protein that is produced by T lymphocytes. Its normal function in the body is to increase the growth and activity of other white blood cells (T and B lymphocytes). When IL-2 is used for cancer therapy, it is manufactured into a product called aldesleukin; a drug used to boost the immune system to fight cancer cells [48]. IL-2 alone or in combination with other anti-cancer therapies has brought some survival benefits to advanced cancer patients. A recent meta-analysis seems to support the use of IL-2 in combination with chemotherapy in solid tumors other than melanoma and renal cancer, reporting a trend toward better prognosis in the response in several solid tumors. Another meta-analysis also shows that IL-2 combination therapy is efficacious in treating non-small cell lung cancer, improves overall survival, and did not show significant toxic reactions. In China, IL-2 has been approved for the treatment of malignant pleural effusion (MPE) since 1998. In subsequent years, some randomized controlled trials (RCTs) have specially explored the clinical efficacy and safety of IL-2 combined with cisplatin versus cisplatin alone in treating MPE through thoracic injection [49].

iii.IL-3:

Interleukin-3 (IL-3) cytokines bind to a receptor made up of a unique IL-3Rα subunit and the common βc subunit. They synergize with other cytokines to stimulate the growth of immature progenitor cells of all lineages, and are therefore known to be a multi-lineage colony-stimulating factor (CSF) [50]. They are equally known to be a multipotent hematopoietic growth factor which is produced by activated T-cells, monocytes/macrophages, and stroma cells. Human IL-3 genes are shown to be located on chromosome 5 near segment 5q31 [51]. Interleukin 3 is an interleukin, a type of biological signal that can improve the body’s natural immune response to disease as part of the immune system. IL-3 works to regulate the inflammatory response in order to clear pathogens by changing the abundance of various cell populations via binding at the interleukin-3 receptor [52].

iv.IL-4:

Interleukin-4 (IL-4) is a multifunctional cytokine which plays an important role in immune response regulation and is involved in various processes associated with the development and differentiation of lymphocytes and the regulation of T cell survival [53]. IL-4 is mainly required for lineage-specific differentiation of Th2 cells and regulation of humoral immune responses. In addition to basophils and mast cells, IL-4 is predominantly secreted by Th2 cells via autocrine signaling. It plays multiple roles in the tumor microenvironment and exhibits immune-suppressive and antitumor activities [54]. IL-4 could also promote the proliferation and survival of several cancer cells. It was found to be over expressed by many human tumor types, including malignant glioma, ovarian, lung, breast, pancreatic, colon, and bladder carcinomas, which also overexpress its receptors (IL-4R) [55].

v.IL-5:

Interleukin-5 (IL-5) is mainly produced by T helper-2 (Th2) lymphocytes and group 2 innate lymphoid cells (ILC2). It can increase antibody secretion by promoting the differentiation and growth of B cells and enhance the humoral immune response mediated by Th2 cells. Immunity to tumors is mainly controlled by Th1-mediated cellular immunity. If Th1-Th2 drift occurs, it will lead to an immunosuppressive response and the development of cancer [56]. IL-5 is involved in a number of immune responses, such as helminth infection and allergies. It also plays an important role in innate immunity by maintaining B-1 B cells and mucosal IgA production [57].

vi.IL-6:

Interleukin-6 (IL-6) belongs to a broad class of cytokines involved in the regulation of various homeostatic and pathological processes. These activities range from regulating embryonic development to wound healing, ageing, inflammation, and immunity, including immunity to COVID-19 [58]. When homeostasis is disrupted by infections or tissue injuries, IL-6 is produced immediately and contributes to host defense against such emergent stress through the activation of acute-phase and immune responses. However, dysregulated excessive and persistent synthesis of IL-6 has a pathological effect on acute systemic inflammatory response syndrome and chronic immune-mediated diseases [59].

vii.IL-7:

Interleukin-7 (IL-7) is a multipotent cytokine that maintains the homeostasis of the immune system. It plays a vital role in T-cell development, proliferation, and differentiation, as well as in B-cell maturation through the activation of the IL-7 receptor (IL-7R) [60]. IL-7 presents antitumor effects in tumors such as glioma, melanoma, lymphoma, leukemia, prostate cancer, and glioblastoma. In vivo administration of IL-7 results in a decreased cancer cell growth in murine models. IL-7 can also induce the production of IL-1α, IL-1β, and TNF-α by monocytes and can also help to inhibit melanoma growth [61].

viii.IL-8:

Interleukin-8 (IL-8), alternatively known as CXCL8, is a pro-inflammatory CXC chemokine. CXCL-8 is a chemoattractant factor for myeloid leukocytes that is produced in large quantities by many solid tumors. Levels of IL-8, which can act upon a variety of immune and nonimmune cells, can provide significant information about tumors, including their size and how likely they are to respond to immunotherapy. This is because the IL-8 produced by tumors can promote angiogenesis, recruit immunosuppressive cells like neutrophils and myeloid-derived suppressor cells (MDSCs), and stimulate epithelial-to-mesenchymal transitions, which is a precursor to metastasis [62]. CXCL8 is one of the dominant transcriptional targets of the inflammatory signaling mediated by nuclear factor-*κ*B (NF-*κ*B), which is commonly activated in cancer cells. CXCL8 is a pro-inflammatory chemokine that acts on leukocytes and endothelial cells via their CXCR1 and CXCR2 receptors to promote immune infiltration and angiogenesis, which in turn establishes a venue for cancer cell local invasion, migration, and metastasis. As an angiogenic chemokine, CXCL8 binds with high affinity to both the CXCR1 and CXCR2 receptors, contributing to its function in the cancer microenvironment [63].

ix.IL-10:

Interleukin-10 (IL-10) is a pleiotropic cytokine that plays a fundamental role in modulating inflammation and maintaining cell homeostasis. It primarily acts as an anti-inflammatory cytokine by protecting the body from an uncontrolled immune response, mostly through the Jak1/Tyk2 and STAT3 signaling pathways. On the other hand, IL-10 can also have an immunostimulating function under certain conditions [64]. IL-10 inhibits the secretion of cytokines, such as IL-1, IL-6, IL-12, IL-18, IL-23, and tumor necrosis factor (TNF), and promotes the differentiation of naïve CD4+ T cells into Tregs. It has been shown that IL-10 has a pivotal role in limiting an excessive inflammatory response, thus reducing collateral host damage while clearance of the pathogen is under way. IL-10 also has a beneficial role in limiting autoimmune diseases [65].

x.IL-12:

Interleukin-12 (IL-12) is a potent, pro-inflammatory type 1 cytokine that has long been studied as a potential immunotherapy for cancer [66]. It is involved in both the innate and adaptive immunity that stimulate T and natural killer cell activity and induce interferon gamma production. IL-12 has been identified as a potential immunotherapeutic component for combinatorial cancer treatments [67]. IL-12 can be considered a strong candidate for immunotherapy-based interventions, as it potentiates tumor-specific cytotoxic NK and CD8+ T cells that are largely responsible for tumor cell killing. However, the systemic administration of IL-12 is quite toxic; therefore, alternative methods of IL-12 delivery and/or the activation of T cells by IL-12 are needed [68].

xi.IL-15:

Interleukin-15 (IL-15) is a cytokine that belongs to the interleukin-2 (IL-2) family and is essential for the development, proliferation, and activation of immune cells, including natural killer (NK) cells, T cells, and B cells [69]. In addition to using IL-15 and its derivatives alone in cancer immunotherapy, IL-15 has also been incorporated into many adoptive cell therapies against cancer, specifically in combination with chimeric antigen receptor (CAR) engineering. In many recent studies, researchers have attempted to incorporate IL-15 not only in ex vivo precultures but also by integrating IL-15 and its receptor in CAR engineering [70].

xii.IL-18:

Interleukin-18 (IL-18) is an immunostimulatory cytokine belonging to the IL-1 family. It can regulate both innate and adaptive immune responses through its effects on natural killer (NK) cells, monocytes, dendritic cells, T cells, and B cells. It can equally act synergistically with other pro-inflammatory cytokines to promote interferon-γ (IFN-γ) production by NK cells, T cells, and possibly other cell types. Systemic administration of IL-18 has been shown to result in significant antitumor activity in several preclinical animal models [71].

xiii.IL-21:

Interleukin (IL)-21 is a cytokine produced by activated conventional CD4+ T lymphocytes and natural killer T cells, driving anti-tumor immunity in the skin and kidney. It can equally be identified as pro-inflammatory in many tissues, and it promotes colitis-associated colon cancer [72]. Initially, IL-21 was recognized for its anti-tumor effects in several preclinical tumor models, warranting its currently ongoing clinical development as a cancer immunotherapeutic. More recently, IL-21 has been associated with the development of a panel of autoimmune and inflammatory diseases, and neutralization of IL-21 has been suggested as a potential new therapy [73].

xiv.IL-23:

Interleukin-23 (IL-23) has been recently identified as a heterodimer cytokine with components related to the IL-6 family of cytokines [74]. IL-23 promotes the differentiation of Th17 cells, which orchestrates neutrophil-initiated resistance to extracellular bacteria and inflammation [75].

xv.GM-CSF:

Granulocyte macrophage colony-stimulating factor (GM-CSF) is a cytokine that drives the generation of myeloid cell subsets including neutrophils, monocytes, macrophages, and dendritic cells in response to stress, infections, and cancers [76]. GM-CSF was identified as a hematopoietic growth factor that causes granulocyte and macrophage colony formation. It is used for DC generation from bone marrow cells or monocytes in vitro. At steady state, however, GM-CSF-deficient mice exhibit no defects in the development of myeloid cells, with the exception of alveolar macrophages and specific DC subsets in nonlymphoid tissues. Instead, GM-CSF plays a role in tissue inflammation and autoimmune diseases, including rheumatoid arthritis, multiple sclerosis, asthma, psoriasis, and type I diabetes [77].

xvi.IFN-α:

Interferon-α (INF-α) has a number of biological effects, including the inhibition of tumor cell growth through mechanisms that are not well understood. The role of INF-α in the treatment of non-Hodgkin’s lymphomas (NHL) was first investigated in a preclinical model of the AKR/J mouse [78]. Phase II trials involving IFN-α were conducted by the National Cancer Institute in non-Hodgkin lymphoma (NHL) patients. Conflicting results were seen with regards to the impact of IFN-α induction monotherapy and maintenance, and when combined with chemotherapy, on the survival of NHL patients. IFN-α was used in treating low-grade indolent NHLs, where it showed some activity; however, the complete response (CR) and overall response rates were only 10% and 48%, respectively [79].

xvii.IFN-γ:

Interferon-γ (IFN-γ) plays a key role in the activation of cellular immunity, and subsequently, the stimulation of the antitumor immune response. Based on its cytostatic, pro-apoptotic, and antiproliferative functions, IFN-γ is considered potentially useful for adjuvant immunotherapy for different types of cancers. Moreover, IFN-γ may inhibit angiogenesis in tumor tissue, induce regulatory T-cell apoptosis, and/or stimulate the activity of M1 proinflammatory macrophages to overcome tumor progression [80].

xviii.TGF-β:

The transforming growth factor beta (TGF-β) cytokine has a research history of more than 40 years. TGF-β is secreted by many tumor cells and is associated with tumor growth and cancer immunity. The canonical TGF-β signaling pathway, SMAD, controls both tumor metastasis and immune regulation, thereby regulating cancer immunity. TGF-β regulates multiple types of immune cells in the tumor microenvironment, including T cells, natural killer (NK) cells, and macrophages. One of the main roles of TGF-β in the tumor microenvironment is the generation of regulatory T cells, which contribute to the suppression of anti-tumor immunity [81]. TGF-β is necessary for lung organogenesis and homeostasis, as evidenced by genetically engineered mouse models. TGF-β is crucial for epithelial–mesenchymal interactions during lung branching morphogenesis and alveolarization. Expression and activation of the three TGF-β ligand isoforms in the lungs are temporally and spatially regulated by multiple mechanisms [82].

## 4. Conclusions and Future Directions

Cytokines have been shown to be effective in the treatment of cancer, but it is not yet clear how some effective targets affect the immune system differently. For cell populations that have been extensively studied, such as T cells, the labeling and role of surface receptors is well understood, and researchers use the same set of cluster of differentiation (CD) markers to identify the populations at hand using flow cytometry. However, the difficulty of detecting the receptor set increases with the level of the immune system being studied, making it difficult to compare results and identify patterns over a long time period.

By analyzing changes in the activity and expression of surface markers of all different types of immune cells after immunotherapy, evaluation of treatment can be made easier and more consistent. This analysis can also use surface markers of the immune system in cancer patients to predict the effectiveness of the immune system.

It is important to examine the role and surface markers of the immune system in tumors and to pursue cytokine-based immunotherapy, but the main purpose of this article is to provide detailed information about the prospects of cancer cytokine therapy. One of the most important features of cytokines as modulators of the immune system is their pleiotropic effects. Each cytokine has a different effect on the immune system, enabling it to promote both pro-inflammatory and anti-tumor responses.

Therefore, for the future of cytokine-based cancer therapy, it will be important to develop a combination strategy that will enhance the anti-inflammatory effects while inhibiting the tumor-enhancing immune system. High doses of cytokines are required to induce a beneficial response in cancer patients, but doing so leads to many problems, including their short lifespan and toxicity (pro-inflammatory and autoimmune response). New technologies are being developed that improve the targeting of cytokines and alter their pharmacokinetics, such as cellular or other delivery systems based on transporters and drug transfer proteins, voluntarily helping to solve many of the shortcomings of cytokine therapy. According to the latest developments in cancer prevention, these will become the most important part of the treatment when used together with other drugs such as cytokines, anti-inflammatory drugs, oncolytics, or as part of the immune response of dendritic cells (DC) and tumor cells.

## Figures and Tables

**Figure 1 cancers-15-05383-f001:**
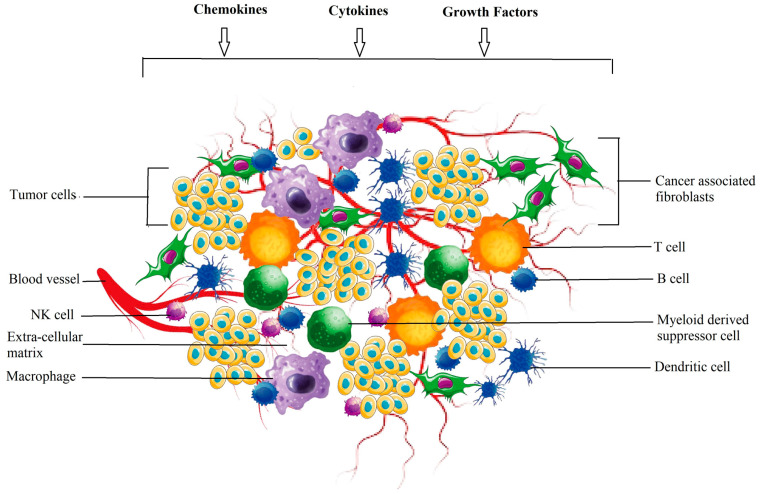
The elements that make up a tumor’s microenvironment. The TME is made up of both cellular and noncellular elements. Cancer-associated fibroblasts (CAFs), adipocytes (APs), niche, immune cells, carcinogenic and noncancerous cells, endothelial cells (ECs), mesenchymal stem cells (MSCs), and mesenchymal stem cells (MSCs) are all present in the former and all contribute to the tumor’s progression. The mediators that make up the noncellular component include growth factors, cytokines, and chemokines. They could develop independently or promote a cancer cell’s growth [20].

**Table 1 cancers-15-05383-t001:** Basic characteristics of cytokines [26].

Cytokine	Primary Cell Source	Primary Target Cell	Biological Activity
IL-1	Monocytes Macrophages Fibroblasts Epithelial cells Endothelial cells Astrocytes	T cells B cells Endothelial cells Hypothalamus Liver	Co-stimulation Cell activation Inflammation Fever Acute phase reactant
IL-2	T cells NK cells	T cells NK cells B cells Monocytes	Cell growth Cell activation
IL-3	T cells	Bone marrow progenitor cells	Cell growth and cell differentiation
IL-4	T cells	T cells B cells	Th2 differentiation Cell growth Cell activation IgE isotype switching
IL-5	T cells	B cells Eosinophils	Cell growth Cell activation
IL-6	T cells Macrophages Fibroblasts	T cells B cells Liver	Co-stimulation Cell growth Cell activation Acute phase reactant
IL-7	Fibroblasts Bone marrow stromal cells	Immature lymphoid progenitors	T cell survival, proliferation, homeostasis B cell development
IL-8	Macrophages Epithelial cells Platelets	Neutrophils	Activation Chemotaxis
IL-10	Th2 T cells	Macrophages T cells	Inhibits antigen-presenting cells Inhibits cytokine production
IL-12	Macrophages NK cells	T cells	Th1 differentiation
IL-15	Monocytes	T cells NK cells	Cell growth Cell activation NK cell development Blocks apoptosis
IL-18	Macrophages	T cells NK cells B cells	Cell growth Cell activation Inflammation
IL-21	CD4+ T cells NKT cells	NK cells T cells B cells	Cell growth/ activation Control of allergic responses and viral infections
IL-23	Antigen-presenting cells	T cells NK cells DC	Chronic inflammation Promotes Th17 cells
GM-CSF	Fibroblasts Mast cells T cells Macrophages Endothelial cells	DC Macrophages NKT cells Bone marrow progenitor cells	T-cell homeostasis Promotes antigen presentation Hematopoietic cell growth factor
IFN-α	Plasmacytoid DC NK cells T cells B cells Macrophages Fibroblasts Endothelial cells Osteoblasts	Macrophages NK cells	Anti-viral Enhances MHC expression
IFN-γ	T cells NK cells NKT cells	Monocytes Macrophages Endothelial cells Tissue cells	Cell growth/activation Enhances MHC expression
TGF-β	T cells Macrophages	T cells	Inhibits cell growth/activation
TNF-α	Macrophages T cells	T cells B cells Endothelial cells Hypothalamus Liver	Co-stimulation Cell activation Inflammation Fever Acute phase reactant
IL-17	NKT cells	Epithelial cells	Control of infections
	ILC	Endothelial cells	Initiate a potent inflammatory response
		Fibroblasts	
		Osteoblasts	
IL-27	Monocytes	T cells	Enhance the proliferation of naïve CD8+ T cells
	DC	NK cells	Stimulate human monocyte to express TLR4
IL-35	T cells	Tregs	Suppresses T-cell proliferation
			Produces iTr35
IL-37	Monocytes	Macrophages	-tv regulate excessive inflammatory response
	Dendritic cells	DCs	
		B cells	

**Table 2 cancers-15-05383-t002:** The functions and duties of the cytokine-secreting cells in the tumor microenvironment [20].

Cell Type	Function in TME
Tumour-associated macrophages (TAMs)	TAMs exhibit the M2 macrophage phenotype, which includes protumorigenic characteristics, anti-inflammatory properties, and Th2 cytokine secretion. These help cancer cells invade secondary areas and promote angiogenesis.
Cancer-associated fibroblasts (CAFs)	Stromal cell populations that are active support the desmoplastic tumor microenvironment. By releasing cytokines, they can encourage angiogenesis and control tumor-promoting inflammation.
CD4+ T_h_ cells	Th1 and Th2 lineages have been divided. Th1 secretes cytokines that are both pro-inflammatory and anti-tumorigenic, whereas Th2 secretes cytokines that are both pro-inflammatory and tumorigenic.
CD8+ T_c_ cells	Adaptive immune system effector cells that recognize and kill tumor cells by perforin-granzyme-mediated apoptosis.
Mast cells (MCs)	Innate and adaptive immune responses to be produced and maintained. Release substances that encourage endothelial cell development to aid tumor cell angiogenesis.
B cells	Modulators of humoral immunity and secrete cytokines. Alter the Th1:Th2 ratio.
Natural killer (NK) cells	Without antigen presentation, cytotoxic lymphocytes obliterate stressed cells. Through “missing self” activation and “stress-induced” activation, they detect and destroy tumor cells.
Dendritic cells (DCs)	Antigen-presenting cells (APCs) that control the immune system’s adaptive response. They increase vascularization in the TME to encourage angiogenesis.
Neutrophils	N1-type cells are pro-inflammatory, anti-tumorigenic, and release Th1 cytokines.

**Table 3 cancers-15-05383-t003:** Classification of cytokine receptors [26].

Receptor Family	Ligands	Structure and Function
Type I cytokine receptors	IL-2 IL-3 Il-4 IL-5 IL-6 IL-7 IL-9 IL-11 IL-12 IL-13 IL-15 IL-21 IL-23 IL-27 Erythropoietin GM-CSF G-CSF Growth hormone Prolactin Oncostatin M Leukemia inhibitory factor	Composed of multimeric chains. Signals through the JAK-STAT pathway using a common signaling chain. Contains cytokine binding chains.
Type II cytokine receptors	IFN-α/β IFN-γ IL-10 IL-20 IL-22 IL-28	Immunoglobulin-like domains. Uses heterodimer and multimeric chains. Signals through JAK-STAT.
Immunoglobulin superfamily receptors	IL-1 CSF1 c-kit IL-18	Shares homology with immunoglobulin structures.
IL-17 receptor	IL-17 IL-17B IL-17C IL-17D IL-17E IL-17F	
G protein-coupled receptors (GPCRs)	IL-8 CC chemokines CXC chemokine	Function to mediate cell activation and migration.
TGF-β receptors ½	TGF-β	
Tumor necrosis factor receptors (TNFRs)	CD27 CD30 CD40 CD120 Lymphotoxin-β	Function as co-stimulatory and co-inhibitory receptors.

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
