# Peer review of "Cytokine Profile in Lung Cancer Patients: Anti-Tumor and Oncogenic Cytokines"

_cancers, 2023, doi:10.3390/cancers15225383_

Round 1
Reviewer 1 Report
-
1. The English in this review requires significant revision, as many sentences are difficult to understand.
-
2. Numerous statements require citation for support.
-
3. Figure 1 should be redesigned; the current layout is unclear.
The English quality needs significant improvement, and there are several areas where the logic is unclear or content is repetitive.
Author Response
Dear Reviewer thank you very much for taking the time to review this manuscript. Please find the corresponding revisions and corrections highlighted in red in the re-submitted file.

Reviewer 2 Report
In this review, the authors outline the types of lung cancer, the tumor microenvironment, basic cytokine properties, cytokine receptors, and the role of cytokines in immunotherapy and cancer.
This includes the topics of cancer development and novel cancer targets, which may provide relevant novel information. However, there are many aspects that should be reconsidered.
The references are not listed in order. Reference numbers 44-76 do not appear in the text. Many other references are unrelated to their position in the text.
Page 8: Figure 3, which is described in the paper, is missing.
The main problem is that Figure 1 and Table 2 approximate Figure 1 and Table 1 in the paper published by Ramachandran et al. (Ref. 79). This must be corrected.
Table 1: IL-17 (page 10), IL-27 (page 11), Il-35 (page 10), IL-37 (ape 11) are listed in the text but not in Table 1.
Tables 1, 2, and 3 need references.
Page 8, 3. The role of cytokines in immunotherapy and cancer: In this section, the authors describe the role of cytokines. Each cytokine needs to be discussed after adding a subtitle such as IL-6, TNF, IL-10, etc. The IL-17 part (pages 9 and 11) needs to be integrated.
Other relevant references need to be added. For example,
1) Cancer Immunol Immunother. 2021 Jul;70(7):1867-1876. doi: 10.1007/s00262-020-02798-z.
2) Cells. 2021 Jan 8;10(1):100. doi: 10.3390/cells10010100.
3) J Clin Invest. 2020 Jul 1;130(7):3560-3575. doi: 10.1172/JCI124037.
4) Cells. 2022 Jul 21;11(14):2257. doi: 10.3390/cells11142257.
5) BMC Med. 2022 May 13;20(1):187. doi: 10.1186/s12916-022-02356-7.
6) Nat Commun. 2022 Oct 15;13(1):6095. doi: 10.1038/s41467-022-33719-6.
7) Qiao M, et al. Interleukin-10 induces expression of CD39 on CD8+ T cells to potentiate anti-PD-1 efficacy in EGFR-mutated non-small cell lung cancer. J Immunother Cancer. 2022 Dec;10(12):e005436. doi: 10.1136/jitc-2022-005436.
can be improved.
Author Response
Dear Reviewer, thank you very much for taking the time to review this manuscript. Please find the corresponding revisions and corrections highlighted in red in the re-submitted file.

Reviewer 3 Report
Dear Authors!
I found your manuscript very interesting and I think that it is suitable for publication after a minor revision.
I suggest to create subdivisions in the third section of your review.
I found the quality of English language very good.
Author Response

(The authors gave the same response as above.)

Reviewer 4 Report
Title: Cytokine profile in lung cancer patients: Anti-tumor and onco-genic cytokines
ID: cancers-2593402
Article Type: Review
Journal: Cancers
Section: Cancer Immunology and Immunotherapy
The aim of the present review was to describe the role of the tumor microenvironment and its components in lung cancer. Furthermore, the authors focused on the biological activity of various cytokines.
Generally, the review is quite complete, but it is very lacking in bibliography that absolutely must be integrated, there are extra spaces or no spaces along the text. Figures mentioned in the text are missing and on the contrary, there are tables not mentioned in the text.
Abstract
I would replace the sentence “Lung cancer accounts for the majority of cancer cases” with “Lung cancer is currently the second leading cause of cancer worldwide”. The sentence is misleading for the reader, in fact lung cancer is not the most common in the world and you state this correctly in the introduction.
Avoid including acronyms in the abstract. Place acronyms such as TMJ or ICI in parentheses in the introduction the first time you introduce the full term.
Introduction
Please authors write CT in full and write CT in brackets if it is mentioned also after in the text.
Bibliography is missing throughout the paragraph. Please add the references.
Every time you cite a work you must include the reference. Please add the reference of the study of R. Kuner et al.
1.1. Type and treatment of lung cancer
Please authors replace “This” with “Lung cancer”. I think it is best to start a paragraph by explaining the subject.
Please authors check all references numbering. There cannot be reference number 70 after number 2 and before number 9.
Please authors write Food and Drug Administration before FDA and write FDA in brackets.
1.2. Tumor microenvironment and content: their contribution to NSCLC metastasis
This paragraph is totally devoid of bibliography. Please authors add the references.
Figure 1
I would propose that the authors modify some colors to better distinguish the different components of the tumor microenvironment. In particular, distinguish chemokines, cytokines and growth factors with 3 different colors and differentiate dendritic cells and B cells with 2 different colors. Furthermore, the tumor cells and cancer associated fibroblasts also appear to be the same color.
2. Basic properties of cytokines
Here too the bibliography is missing. Please authors to insert it also in the tables.
Please authors to insert the missing acronyms in the tables.
Table 2 is not mentioned in the text. Please do this before its inclusion in the manuscript.
2.1. Classification of cytokines and their receptors
Here too the bibliography is missing. Please authors to insert it also in the table.
2.1.1. Type I cytokine receptors
I read in this paragraph “Figure 3” but I see only one figure in the work (Figure 1). Why?
2.1.3. Immunoglobulin superfamily receptors
Here too the bibliography is missing. Please authors to insert the references.
3. The role of cytokines in immunotherapy and cancer
Please authors to insert the references at the end of each sentence. Furthermore, please authors check all references numbering.
In this paragraph, the authors could insert information from this article: Defense and protection mechanisms in lung exposed to asbestiform fiber: The role of macrophage migration inhibitory factor and heme oxygenase-1 by Loreto et al. (DOI 10.4081/ejh.2020.3073).
4. Conclusions and future directions
Please authors write CD in full and write CD in brackets. Same goes for DC etc.
Author Response

(The authors gave the same response as above.)

Round 2
Reviewer 1 Report
Thank you for the insightful review paper. To enhance its relevance, it would be beneficial to delve into the cytokine profiles associated with different treatment modalities in lung cancer.
Author Response
Dear Reviewer, Thank you very much for taking the time to review this manuscript. Please find the corresponding revisions highlighted in red in the re-submitted file.

Reviewer 2 Report
The authors respond appropriately to the points raised by the reviewer.
Author Response
Dear Reviewer, Thank you very much for taking the time to review this manuscript.
Reviewer 4 Report
Title: Cytokine profile in lung cancer patients: Anti-tumor and onco-genic cytokines
ID: cancers-2593402
Article Type: Review
Journal: Cancers
Section: Cancer Immunology and Immunotherapy
The aim of the present review was to describe the role of the tumor microenvironment and its components in lung cancer. Furthermore, the authors focused on the biological activity of various cytokines.
Abstract
The second time the authors write TMJ you do not even have to write it in full.
Introduction
Every time you cite a work you must include the reference. Please add the reference of the study of R. Kuner et al.
1.1. Lung cancer
Please authors replace “This” with “Lung cancer”. I think it is best to start a paragraph by explaining the subject.
Please authors write Food and Drug Administration before FDA and write FDA in brackets.
3. The role of cytokines in immunotherapy and cancer
In this paragraph, the authors could insert information from this article: Defense and protection mechanisms in lung exposed to asbestiform fiber: The role of macrophage migration inhibitory factor and heme oxygenase-1 by Loreto et al. (DOI 10.4081/ejh.2020.3073).
Author Response

(The authors gave the same response as above.)
